# Engineering shape memory and morphing protein hydrogels based on protein unfolding and folding

Qingyuan Bian[1], Linglan Fu[1] & Hongbin Li [1✉]

Engineering shape memory/morphing materials have achieved considerable progress in polymer-based systems with broad potential applications. However, engineering protein-based shape memory/morphing materials remains challenging and under-explored. Here we report the design of a bilayer protein-based shape memory/morphing hydrogel based on protein folding-unfolding mechanism. We fabricate the protein-bilayer structure using two tandem modular elastomeric proteins (GB1)$_8$ and (FL)$_8$. Both protein layers display distinct denaturant-dependent swelling profiles and Young's moduli. Due to such protein unfolding-folding induced changes in swelling, the bilayer hydrogels display highly tunable and reversible bidirectional bending deformation depending upon the denaturant concentration and layer geometry. Based on these programmable and reversible bending behaviors, we further utilize the protein-bilayer structure as hinge to realize one-dimensional to two-dimensional and two-dimensional to three-dimensional folding transformations of patterned hydrogels. The present work will offer new inspirations for the design and fabrication of novel shape morphing materials.

---

[1] Department of Chemistry, University of British Columbia, Vancouver, BC V6T 1Z1, Canada. ✉email: Hongbin@chem.ubc.ca

Hydrogels are three-dimensional hydrophilic polymer matrices that have high water-absorbent capacities[1,2]. Hydrogels exhibit excellent physicochemical features, such as high specific surface areas, thermal insulation, permeability as well as designable mechanical and optical properties[1,3]. Therefore, they have found a wide range of applications including biosensors[4], tissue engineering scaffolds[5], microfluidic devices[6], drug delivery vehicles[7], and etc.

Traditional hydrogels are relatively inert and static. In contrast, biological systems are dynamic and can change their properties by sensing and responding to environmental changes[8]. To mimic nature's dynamic sensing-responding process, smart stimulus-responsive hydrogels have gained increasing interest. These dynamic hydrogels can change their volume, pore size, mechanical properties, optical transparency, or undergo solution–gel transitions in response to environmental changes, such as temperature, pH, magnetic field, redox potential, light, and shear stress[8,9]. Shape-morphing materials that can undergo out-of-plane deformations in response to stimuli have shown promising applications in microfluidic switches[10], artificial muscles[11], and soft robotics[12]. Dynamic hydrogels offer some unique advantages in engineering such shape-morphing materials, including their aqueous nature, large volume changes, as well as responsiveness to a range of different physical stimuli and biochemical signals[13].

However, hydrogels are typically isotropic and only exhibit uniform volumetric expansion or contraction under stimuli[14]. Hence, several strategies have been developed to create heterogeneous hydrogel structures, such as introducing a gradient distribution of responsive components through the hydrogel[15] and combining different materials into a multilayer construct[16]. Due to the flexibility to design individual layer with different geometries and properties, bilayer hydrogels whose two layers exhibit asymmetric responsive behaviors are promising to achieve fast, sensitive, and tunable shape morphing. Many smart bilayer hydrogels composed of synthetic polymers[17], DNAs[18], natural polysaccharides[19], or their hybrids have been developed, and temperature and solvent conditions are the most frequently used stimuli.

Over the last two decades, recombinantly engineered proteins have also become attractive building blocks for engineering hydrogels, thanks to the development of recombinant DNA technology and protein engineering techniques, which allows for precise control over the sequence, folded structure, chain length, and mono-dispersity of the resultant proteins[9,20,21]. In addition, it becomes possible to engineer proteins that contain different functional and structural domains, leading to easier control and prediction of the biological, physical, and mechanical properties of the final protein hydrogels[22]. Despite the significant progress in engineering functional protein-based hydrogels, developing protein-based shape memory/morphing hydrogels remains largely under-explored[23,24]. Recently, Popa and coworkers explored the use of folded globular protein bovine serum albumin to engineer shape memory protein hydrogels by employing polymer reinforcements[23] and metal chelation strategies[24]. The thermoresponsive behaviors of elastin-like polypeptide, and temperature or salt-induced helix to coil transition of gelatin were also used to engineer shape-memory protein hydrogels[25–27]. Here, we report the feasibility of using protein folding–unfolding as a mechanism to engineer shape-morphing protein hydrogels based on the bilayer architecture.

## Results and discussion

**Design principles to engineer shape-morphing protein hydrogels based on protein folding–unfolding.** Bilayer shape-morphing hydrogels are based on the asymmetric responsive behaviors of the two hydrogel layers, due to their different swelling and mechanical properties. Thus, it should be feasible to design shape-morphing protein hydrogels by using two protein hydrogels, if their mechanical and swelling properties can be regulated by external stimuli. Since protein unfolding-folding can lead to significant changes of the mechanical and swelling properties of elastomeric proteins-based hydrogels, we hypothesize that protein folding-unfolding, the ultimate conformational changes of proteins, can be used as a mechanism to program

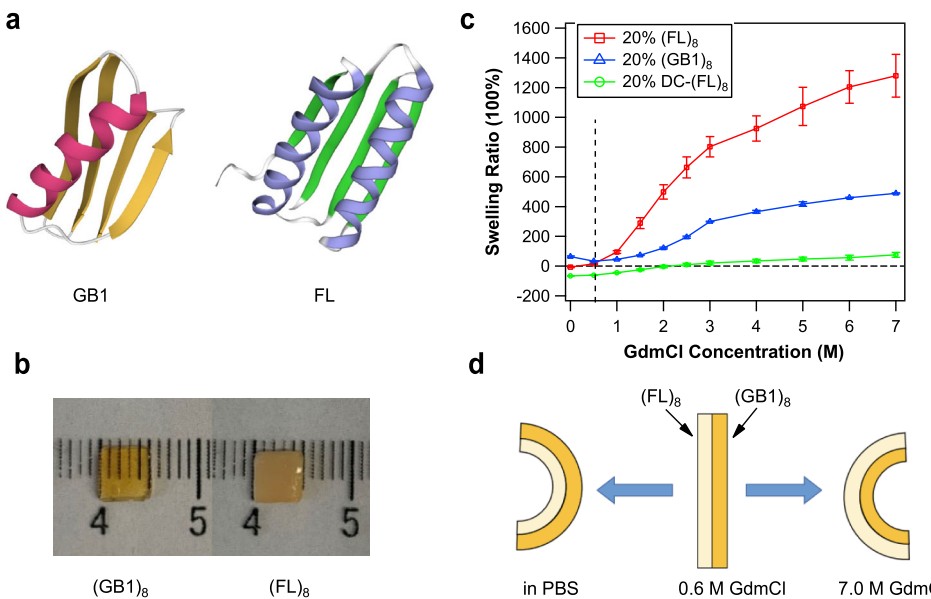

**Fig. 1 Engineering protein-based shape-morphing hydrogels based on bilayer structures. a** Left: The structure of folded globular GB1 domain (PDB code 1PGA). Right: The modeled three-dimensional structure of the FL domain using SWISS MODEL. **b** (GB1)$_8$ (left) and (FL)$_8$ (right) could be readily gelated using the [Ru(bpy)$_3$]$^{2+}$-mediated photochemical cross-linking strategy. **c** Swelling ratios (SR) of 20% (FL)$_8$, 20% (GB1)$_8$, and 20% denatured-cross-linking-(FL)$_8$ (DC-(FL)$_8$) hydrogels at different GdmCl concentrations. The SR of 20% (FL)$_8$ exceeded that of 20% (GB1)$_8$ hydrogel at around 0.6 M [GdmCl], as indicated by the dashed line. **d** The predicted bidirectional bending behaviors of (GB1)$_8$/(FL)$_8$ bilayer hydrogel in response to GdmCl concentration change.

shape-morphing properties of bilayer protein hydrogels in a controlled and predictable fashion. Protein folding and unfolding are characterized by their equilibrium chemical unfolding curves, which reflect the response of proteins to the chemical denaturants. Since different proteins exhibit different responses to chemical denaturants, we reason that choosing two proteins with different chemical denaturation characteristics to construct bilayer protein hydrogels should allow us to engineer shape-morphing protein hydrogels by using chemical unfolding to induce asymmetric changes in their swelling behaviors.

As a proof-of-principle, here we chose two well-studied protein hydrogels based on $(GB1)_8$ and $(FL)_8$ to demonstrate the feasibility to engineer shape-morphing protein hydrogels based on protein folding and unfolding. $(GB1)_8$ and $(FL)_8$ are two tandem modular elastomeric proteins that contain eight tandem repeats of small globular proteins GB1 and FL (Fig. 1a), respectively. Their mechanical properties have been well-studied previously[28,29]. FL is a computationally redesigned variant of the designer protein Di-I_5[30], while GB1 is the B1 immunoglobulin-binding domain of protein G of streptococcal[31]. Both proteins can be readily cross-linked into protein hydrogels by using the well-developed $[Ru(bpy)_3]^{2+}$-mediated photochemical cross-linking strategy[29,32,33]. In addition, FL and GB1 show different chemical denaturation characteristics (Supplementary Fig. 1). The midpoints of the chemical denaturation $[D]_{0.5}$, at which 50% of the proteins are unfolded, are 1.5 M and 2.8 M for FL[29] and GB1[34], respectively, making it possible to use chemical denaturant to induce asymmetric swelling response of the bilayer hydrogel constructed from $(GB1)_8$ and $(FL)_8$.

**Chemical denaturant induced different swelling responses of the $(GB1)_8$ and $(FL)_8$ hydrogels**. We first carried out experiments to confirm that the different equilibrium unfolding characteristics of GB1 and FL can lead to different swelling properties of $(GB1)_8$ and $(FL)_8$ hydrogels. Figure 1b shows the photographs of the two protein hydrogels at a protein concentration of 20%. The obtained $(GB1)_8$ hydrogel was transparent while the $(FL)_8$ hydrogel was opaque. The two hydrogels showed different swelling behaviors in PBS (Fig. 1c): the $(GB1)_8$ hydrogel swelled in PBS with a swelling ratio (SR) of $64 \pm 5\%$ (average ± standard deviation, $n = 6$), while the $(FL)_8$ hydrogel deswelled showing a negative SR of $-7 \pm 3\%$ ($n = 6$).

The equilibrium-swelling properties of polymer hydrogels can be described by the classical Flory–Rehner theory of network swelling[35,36]. During swelling, the equilibrium is reached when the elastic free energy of the protein network balances the mixing free energy. The swelling ratio of $(GB1)_8$ network (~64%) behaves as a typical polymer hydrogel made of hydrophilic polymers. In contrast, the deswelling behavior of the $(FL)_8$ is unusual. As we demonstrated before[29], FL is mechanically labile and unfolds at ~5 pN. Upon swelling of the hydrogel network, the swelling force is sufficient to trigger the unfolding of a fraction of FL domains in the hydrogel network, generating mechano-chemical coupling[37–39]. Due to the fact that PBS is a poor solvent of unfolded protein, unfolded FL polypeptide chains underwent hydrophobic collapse and aggregation, leading to the deswelling and opaque appearance[29].

When $(GB1)_8$ and $(FL)_8$ hydrogels were transferred from PBS to GdmCl-containing buffers, the swelling ratios of both hydrogels increased with the increasing [GdmCl] (from 0 M to 7.0 M) (Fig. 1c), largely following the trend of the chemical denaturation curves. The SR of $(GB1)_8$ increased from 64% in PBS to $489 \pm 3\%$ in 7 M GdmCl ($n = 6$). In contrast, $(FL)_8$ hydrogel displayed a much more significant increase in its SR in GdmCl: its SR increased from $-7\%$ in PBS to $1280 \pm 144\%$

($n = 6$) in 7 M GdmCl. It is worth noting that the SR of $(FL)_8$ and $(GB1)_8$ hydrogels became equal at ~0.6 M GdmCl. When [GdmCl] was greater than 0.6 M, the SR of $(FL)_8$ hydrogel surpassed that of $(GB1)_8$, and the SR difference between the two hydrogels increased with the increase of [GdmCl]. The slight decrease in the swelling degree of $(GB1)_8$ hydrogel at 0.5 M GdmCl solution was likely a result of the ionic strength change in the aqueous environment when transferred from PBS to 0.5 M GdmCl.

Based on these results, we predicted that the bilayer hydrogel consisting of $(GB1)_8$ and $(FL)_8$ layers would display a bidirectional bending behavior in response to GdmCl concentration change and that the bending angle would increase with increasing denaturant concentration, as schematically shown in Fig. 1d.

**$(GB1)_8/(FL)_8$ bilayer hydrogels exhibited shape morphing in response to PBS and GdmCl**. To test our design principles of engineering shape-morphing hydrogels based on protein unfolding-folding, we fabricated $(GB1)_8/(FL)_8$ bilayer hydrogel strips with a thickness of 1.2 mm by using a 3D-printed mold. We first constructed $(FL)_8$ hydrogel layer using the $[Ru(bpy)_3]^{2+}$-mediated photo-cross-linking method[32], and then formed the $(GB1)_8$ hydrogel layer on top of the $(FL)_8$ layer using the same photo-cross-linking method. Since both hydrogel layers were constructed using the same cross-linking strategy, the two layers were bounded together covalently to form one hydrogel strip. The bounding was strong, and no delamination between the two hydrogel layers was observed in swelling and even tensile testing, making it possible to entail shape morphing.

As shown in Fig. 2a, the bilayer hydrogel retained its straight strip shape after it was taken out of the mold. A clear boundary between the two protein layers was visible. Once immersed in PBS, the bilayer strips spontaneously self-bent towards the $(FL)_8$ layer side and reached a bending angle of $95 \pm 14°$ ($n = 3$) after the swelling reached equilibrium in about half an hour (Fig. 2b, c and Supplementary Fig. 2). When the GdmCl concentration was adjusted to 0.6 M, the hydrogel bending angle decreased to nearly zero, indicating that an equal SR of the $(GB1)_8$ and $(FL)_8$ hydrogel layers was achieved. When the GdmCl concentration was further increased, the hydrogel bending angle became negative ($-65 \pm 13°$ in 7.0 M GdmCl, $n = 3$) as the bilayer strips bent toward the $(GB1)_8$ layer side. Since the swelling/deswelling is mainly controlled by the diffusion of solvent in and out of protein hydrogels, swelling/deswelling are relatively slow processes and take about 1 h in PBS and 2 h in 7.0 M GdmCl to complete (Supplementary Fig. 2)[38].

This bidirectional bending behavior of $(GB1)_8/(FL)_8$ bilayer hydrogel strips in response to GdmCl concentration change was consistent with our prediction based on the swelling properties of each protein layer. When equilibrated in PBS buffer, $(GB1)_8$ layer swelled while $(FL)_8$ layer shrank, which synergistically drove the hydrogel to bend towards the $(FL)_8$ layer side. When the concentration of GdmCl was increased to >0.6 M, $(FL)_8$ layer had a much higher SR than $(GB1)_8$ layer. The significant difference in swelling generated enough driving force to bend the bilayer strip toward the less swollen $(GB1)_8$ layer side[20]. These results clearly demonstrated the feasibility of using protein unfolding-folding to program shape-morphing properties into protein hydrogels.

As illustrated in Fig. 2c, an increase in the hydrogel absolute bending angle was observed when [GdmCl] increased to 2.0 M. This was due to the increased disparity in the swelling ratios of the two protein layers, which generated a larger bending moment. However, upon a further increase in [GdmCl] (from 2.0 to 7.0 M), the bending angle gradually reached an asymptotic value,

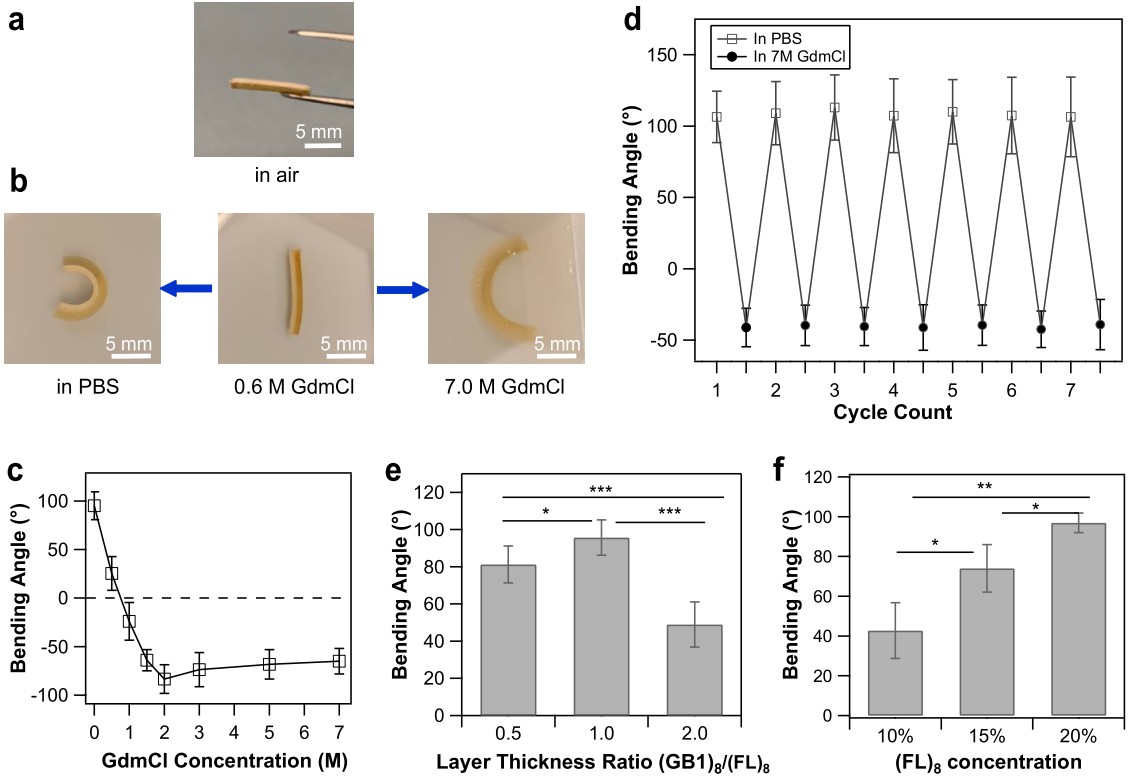

**Fig. 2 Protein-bilayer hydrogel strips constructed from (GB1)$_8$ and (FL)$_8$ display shape-morphing behaviors. a** Linear (GB1)$_8$/(FL)$_8$ bilayer strip in air with a clear boundary between the two protein layers (upper layer: (FL)$_8$, lower layer: (GB1)$_8$). **b** The observed bidirectional bending deformations of the as-prepared (GB1)$_8$/(FL)$_8$ bilayer strip. The bilayer strip bent towards the (FL)$_8$ side in PBS and bent toward the (GB1)$_8$ layer side when GdmCl concentration was above 0.6 M. **c** The variation of (GB1)$_8$/(FL)$_8$ bilayer strip bending angle with the environmental GdmCl concentration ($n = 3$). The dimension of the hydrogel strip is 12 mm in length and 3 mm in width. **d** Reversible bending of (GB1)$_8$/(FL)$_8$ bilayer hydrogels with the environment switched between in PBS and in 7.0 M GdmCl for seven cycles ($n = 3$). **e** The bending angle of (GB1)$_8$/(FL)$_8$ bilayer hydrogels varied with the layer thickness ratio ($n = 5$). **f** The bending angle of (GB1)$_8$/(FL)$_8$ bilayer hydrogels varied with the protein concentration of the (FL)$_8$ layer. (FL)$_8$ concentration was varied from 10 to 20% while (GB1)$_8$ concentration was fixed at 20% ($n = 3$). Statistical significance was evaluated by using the Student $t$ test. Significance levels indicated by asterisks: ***$P < 0.001$, **$P < 0.01$, *$P < 0.05$. The error bars in (**c–f**) are standard deviations.

despite the increased swelling ratio difference between the two hydrogel layers. This could be explained by the different dependencies of bending stiffness and bending moment on the total thickness of the bilayer strip. Generally, bending stiffness increases cubically with material thickness, while bending moment only has a quadratic dependency on the thickness[16]. When the GdmCl concentration was smaller than 2.0 M, the increase in bilayer thickness due to hydrogel swelling was relatively small and the induced bending moment was larger than the bending stiffness. However, at higher GdmCl concentrations, the bending stiffness increased rapidly due to the significant increase in hydrogel thickness, preventing the further increase in bending degree.

**Tunable and reversible deformation behaviors of (GB1)$_8$/(FL)$_8$ bilayer hydrogels.** Since protein folding-unfolding are reversible, we expect that the shape morphing induced by protein folding-unfolding should be reversible. Indeed, as shown in Fig. 2d, the PBS- and GdmCl-responsive bending behaviors of the bilayer hydrogels exhibited excellent reversibility and repeatability. (GB1)$_8$/(FL)$_8$ bilayer hydrogels were alternately soaked in PBS and 7.0 M GdmCl solution for seven cycles. Thanks to the high reversibility of protein folding/unfolding of GB1[37,40] and FL[29,38], the resulted swelling property changes of each protein hydrogel layer were fully reversible, so was the bending angle in response to the change of PBS and GdmCl buffers. Besides, there was no noticeable delamination between the two protein layers despite

repeated deformation, indicating the strong interfacial bonding ability of the two layers. Similar reversible morphing behaviors were also observed in polyelectrolyte-reinforced bovine serum albumin-based hydrogels when immersed in GdmCl and Tris buffers sequentially[23].

The bending behaviors of the bilayer hydrogel depend on the layer geometries and swelling/mechanical properties of the constituent hydrogels. According to the Timoshenko's bilayer theory (Eq. (1)) developed for bimetallic bilayer beams, the curvature of a bilayer strip depends on the stiffness ratio and thickness ratio of the two layers[41,42].

$$k = \frac{\triangle\alpha}{h} \times \frac{6(1 + m)^2}{3(1 + m)^2 + (1 + mn)(m^2 + \frac{1}{mn})} \quad (1)$$

where $k$ is bilayer curvature, $h$ is total thickness, $\triangle\alpha$ is the mismatch in the thermal expansion coefficients, $n$ is the stiffness ratio of both layers and $m$ is the thickness ratio of both layers[42]. For bilayer hydrogel strips, $\triangle\alpha$ is the mismatch between the swelling equilibrium ratios of the two hydrogel layers[43].

To study the effects of layer geometries on bending behaviors, bilayer hydrogels with various layer thickness ratios were prepared while the total thickness was fixed at 1.2 mm. As shown in Fig. 2e. (GB1)$_8$/(FL)$_8$ bilayer hydrogels exhibited the largest bending angle in PBS when the thickness ratio was 1 (with the total thickness constant). This result was qualitatively in agreement with the predicted trend by using the Timoshenko's theory based on the Young's moduli of 20% (GB1)$_8$ and 20%

(FL)$_8$ hydrogels in PBS[29,33], and the thickness ratios of the two hydrogel layers. However, a quantitative agreement between experimental results and theory is still lacking. This is similar to previous studies on hydrogel or polymer bilayer strips[16,42,44]. In addition, the bending angle of the bilayer hydrogel decreased with the increase of the total thickness, an observation that was in line with the prediction by Eq. (1) that the bilayer curvature is inversely proportional to the total thickness (Supplementary Fig. 3).

Since protein concentration can directly affect the Young's modulus and SR of the protein hydrogels, we also investigated the effect of protein concentration on the bending angle of the (GB1)$_8$/(FL)$_8$-based bilayer hydrogels. As shown in Supplementary Table 1, the Young's moduli of (FL)$_8$ hydrogels increased while the swelling ratio decreased as the increase of protein concentration. As a result, we observed that increasing the protein concentration of the (FL)$_8$ hydrogel layer while keeping (GB1)$_8$ concentration at 20% led to the increased curvature (Fig. 2f) in PBS, qualitatively in agreement with Eq. (1).

**Denatured cross-linking hydrogels provide a facile method to program shape morphing of protein hydrogels.** Using (GB1)$_8$ and (FL)$_8$ hydrogels as model systems, we have demonstrated that protein unfolding/folding can be used as an effective means to introduce shape morphing in protein hydrogels. Our results showed that the Timoshenko's bilayer theory can be used as a general guideline to rationally tune the bending angles of bilayer hydrogel strips by varying geometry and swelling/mechanical properties of the constituent hydrogels. In particular, tuning the mismatch between the swelling equilibrium ratios/mechanical properties of the two hydrogel layers will allow for effective controlling the degree of morphing of the bilayer hydrogel strips.

Hence, to further expand the degree of morphing, we intended to further increase the mismatch between swelling/mechanical properties of the two hydrogel layers. For this, we took advantage of the recently developed denatured cross-linking (DC) protein hydrogel strategy[45].

Different from traditional protein hydrogels, which are cross-linked into hydrogels in their native state, DC protein hydrogels are prepared by cross-linking proteins in their unfolded state[45]. Due to the presence of physical entanglement of unfolded protein chains in the concentrated unfolded protein solution, cross-linking unfolded proteins resulted in DC hydrogels with significantly lower swelling and higher Young's modulus in both renatured and denatured hydrogels. As shown in Fig. 1c, the DC-(FL)$_8$ hydrogels which were prepared in 7.0 M GdmCl exhibited considerably lower SR than (GB1)$_8$ and (FL)$_8$ hydrogels in the tested range of [GdmCl] (0–7.0 M), entailing a potential larger morphing degree in bilayer hydrogels incorporating such DC-(FL)$_8$ hydrogels.

To use the DC hydrogel approach to engineer bilayer protein hydrogels, we first constructed the (FL)$_8$ layer using the DC cross-linking approach, in which (FL)$_8$ solution with a given GdmCl concentration was photochemically cross-linked. Then the (GB1)$_8$ layer was prepared on top by using the regular cross-linking approach (i.e., cross-linking (GB1)$_8$ in PBS). Indeed, such a bilayered protein hydrogel displayed a much larger bending angle of 151 ± 15° ($n = 4$) when the hydrogel was equilibrated in PBS, leading to a full-moon circular shape (Fig. 3a). It is of note that the swelling and deswelling process of (GB1)$_8$/DC-(FL)$_8$ took about 30 min. to complete (Supplementary Fig. 4), faster than that of (GB1)$_8$/(FL)$_8$.

Soaking the hydrogel in buffers containing GdmCl resulted in further morphing of the bilayer hydrogel (Fig. 3b and Supplementary Fig. 4). With the increase of [GdmCl], the bending angle

of (GB1)$_8$/DC-(FL)$_8$ bilayer hydrogel first decreased, then slightly increased and reached a plateau (Fig. 3b). This trend was in good agreement with the change in SR difference between the DC-(FL)$_8$ and (GB1)$_8$ hydrogel layers (Fig. 1c), and reflected the changes in bending stiffness and binding moment as a function of the hydrogel thickness. It is of note that bilayer hydrogels consisting of (GB1)$_8$ and DC-(FL)$_8$ protein layers only displayed unidirectional bending deformation towards the DC-(FL)$_8$ layer side despite changes of GdmCl concentration in the soaking bath (Fig. 3b), as the SR of DC-(FL)$_8$ is significantly smaller than that of (GB1)$_8$ layer across the entire tested [GdmCl]. Moreover, (GB1)$_8$/DC-(FL)$_8$ bilayer hydrogels also displayed excellent reversibility and stability in their morphing behaviors (Fig. 3c), thanks to the reversibility of the protein folding/unfolding process.

**Programming hydrogel shape morphing using bilayer hinges.** The bending direction and degree of (GB1)$_8$/(FL)$_8$ bilayer hydrogel strips showed strong dependence on the bilayer geometry and denaturant concentration. The excellent tunability of the bilayer bending behaviors as well as the large bending angle offered by the DC-(FL)$_8$ hydrogel provided feasibility for realizing controllable and programmable morphing of protein hydrogels.

As a basic example, we first demonstrated the programming of an 1D strip into distinct 2D architectures. A full-moon circular shape achieved by using (GB1)$_8$/DC-(FL)$_8$ bilayer hydrogel strip (with DC-(FL)$_8$ layer prepared in 7.0 M GdmCl solution) shown in Fig. 3a is the simplest demonstration. By soaking (GB1)$_8$/DC-(FL)$_8$ bilayer hydrogel in buffers containing different [GdmCl], arcs of different radii could be realized readily (Fig. 4a).

More sophisticated hydrogel shape morphing can be realized by localizing bending only to specified positions[14]. The bilayer units can be utilized as local hinges that drive the transformation of the entire hydrogel in response to external stimuli. To prove this concept, we fabricated a strip hydrogel that contained one anisotropic (GB1)$_8$/DC-(FL)$_8$ bilayer unit and two isotropic (GB1)$_8$ units. The (GB1)$_8$/DC-(FL)$_8$ bilayer hinge bent at swelling equilibrium in PBS while the isotropic (GB1)$_8$ regions remained linear, leading to an overall folding deformation (Fig. 4b). The location of folds could be controlled by the deliberate patterning of the two units.

Furthermore, the folding angle could be precisely programmed by tuning the width of the bilayer hinge (Fig. 4c), leading to the feasibility of programming more sophisticated shape-morphing behaviors. To further explore this possibility, we fabricated a strip hydrogel with three (GB1)$_8$ units spaced by two 2 mm (GB1)$_8$/DC-(FL)$_8$ units as shown in Fig. 4d. When soaked in PBS buffer, the straight hydrogel strip with patterned hinge morphed into a triangle within ~30 min. (Supplementary Movie 1). After equilibrated in 7.0 M GdmCl solution, the strip relaxed into a staple shape (Supplementary Movie 2). After it was immersed back in PBS, the triangle shape recovered within ~30 min. as the GdmCl diffused out. In a similar patterning fashion, we designed another two hydrogel strips that could respectively morph into a rectangular shape and rhombic shape at swelling equilibrium in PBS. Both deformations could be relaxed to a U shape in 7.0 M GdmCl solution and reversibly recovered once the hydrogels were equilibrated in PBS again. These morphing behaviors made the patterned hydrogels strips as shape-memory materials.

Further, a cubic shape hydrogel could be built from the (GB1)$_8$/DC-(FL)$_8$ bilayer hinges and (GB1)$_8$ units (Fig. 4e). The bending response of the bilayer units enabled the inactive (GB1)$_8$ units to stand up from under their own weight, indicating that the force generated by the bilayer deformation was strong and could be utilized to realize more complex 3D shape morphing from 2D hydrogel structures.

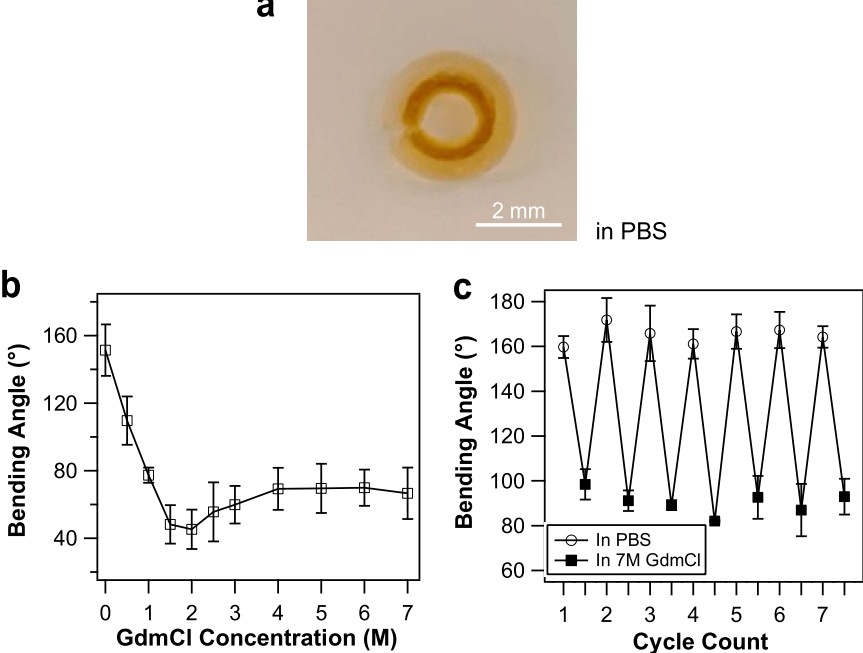

**Fig. 3 Protein-bilayer hydrogel strips prepared by using denature cross-linking (DC) method display increases in the degree of shape morphing.**
**a** $(GB1)_8/DC$-$(FL)_8$ bilayer hydrogel strip achieved full-moon circular shape in PBS. The DC-$(FL)_8$ layer was prepared in 7.0 M GdmCl solution. **b** The effects of GdmCl concentration on the bending angle of $(GB1)_8/DC$-$(FL)_8$ bilayer hydrogels. The DC-$(FL)_8$ layer was prepared in 7.0 M GdmCl solution. **c** Reversible and repeatable bending of $(GB1)_8/DC$-$(FL)_8$ bilayer hydrogels with the environment switched between in PBS and in 7.0 M GdmCl for seven cycles ($n = 4$). Error bars are standard deviations.

**Protein unfolding and folding represent a general approach to engineer protein-based shape-morphing materials.** By taking advantage of the significantly different swelling behaviors of the two protein hydrogels in response to GdmCl, we developed a programmable shape-morphing $(GB1)_8/(FL)_8$ bilayer hydrogel. The bidirectional bending behavior of the bilayer hydrogel was based on the GdmCl-induced asymmetric unfolding of $(GB1)_8$ and $(FL)_8$ proteins. By using a layer-by-layer fabrication strategy, we were able to design the geometry and properties of each protein layer individually, which endowed the resulting $(GB1)_8/(FL)_8$ bilayer hydrogels excellent bending tunability. Such programmable bending/morphing showed excellent reversibility and stability in the programmed shape. 1D to 2D and 2D to 3D hydrogel shape morphing were realized by incorporating $(GB1)_8/DC$-$(FL)_8$ bilayer units as hinges to drive the overall hydrogel transformation. This work demonstrated the feasibility to utilize protein folding-unfolding as a mechanism to engineer shape morphing and shape memory bilayer protein hydrogels. Protein folding and unfolding resulted in significant swelling ratio changes of the individual hydrogel layer, leading to the distinct shape morphing of the bilayer protein hydrogel. Our work demonstrated the potential to use globular protein-based elastomeric proteins as building blocks to engineer shape-morphing hydrogels, where the shape morphing can be precisely programmed by using different proteins and stimuli. Compared with temperature-induced shape morphing and shape memory, protein folding-unfolding provides a general and versatile method to engineer protein-based shape morphing/memory materials, as folding-unfolding is a universal property of folded proteins. Since folding-unfolding can be triggered by using a wide range of stimuli, we expect that many different stimuli can be used to regulate the morphing of protein hydrogels. Therefore, the method we demonstrate here will expand the reservoir of shape-morphing materials and help pave the way of dynamic protein hydrogels for

new applications in areas such as artificial muscles, soft robotics, and biomedical engineering.

## Methods

**Protein engineering.** The amino acid sequence of FL domain is MGEFDIRFRT DDDEQFEKVL KEMNRRARKD AGTVTYTRDG NDFEIRITGI SEQNRKELAK EVERLAKEQN ITVTYTERGS LE. The sequence of GB1 domain is MTYK-LILNGK TLKGETTTEA VDAATAEKVF KQYANDNGVD GEWTYDDATK TFTVTE. Polyproteins $(GB1)_8$ and $(FL)_8$ were previously constructed in our group[24,27] and newly expressed in *Escherichia coli* (*E. coli*) strain DH5α for this work. The bacteria cultures were incubated at 37 °C in 2.5% Luria–Bertani broth containing 100 mg/L ampicillin. Protein overexpression was induced by 1 mM isopropyl-1-β-D-thiogalactopyranoside when the optical density at 600 nm reached 0.6–0.8. Protein expression continued at 37 °C for 4 h. The cells were harvested by centrifugation at 4000 r.p.m. (3488×g) for 10 min at 4 °C and stored at −80 °C overnight. Cell lysis was done in PBS buffer containing protease inhibitor cocktail (0.3% v/v), lysozyme from egg white (1.25 mg/mL), Triton X-100 (10 mg/mL), DNase I (5 μg/mL) and RNase (5 μg/mL). The supernatant containing soluble proteins was collected by centrifugation at 12,000 r.p.m., 4 °C for 1 h. The target proteins carry an N-terminal His-tag and thus were purified by $Co^{2+}$-affinity chromatography (Supplementary Fig. 5). The purified proteins were dialyzed against deionized water at 4 °C for 24 h and then lyophilized.

**Hydrogel preparation and swelling test.** Hydrogels of individual protein were prepared based on a well-developed $[Ru(bpy)_3]^{2+}$-mediated photochemical cross-linking strategy following protocol as described[24,35]. This reaction involves brief photolysis of $[Ru(bpy)_3]^{2+}$ in the presence of a persulfate. The generated Ru(III) and sulfate radical couple two tyrosine residues in close proximity (Supplementary Fig. 6) and covalently crosslink proteins into a network. To prepare 20% $(GB1)_8$ and $(FL)_8$ hydrogels, the pre-gel solutions contained 200 mg/mL of polyprotein, 50 mM ammonium persulfate (APS) and 0.26 mM $[Ru(II)(bpy)_3]Cl_2$ in PBS buffer. 20% DC-$(FL)_8$ hydrogel pre-gel solution was prepared in 7.0 M GdmCl solution with the same reagent composition. The pre-gel solutions were first centrifuged at 12,000 r.p.m. for 4 min to remove trapped air bubbles and then cast into a square-shaped aluminum mold. Irradiation was applied at room temperature using a 200 W fiber optical white light source placed 10 cm above the mold for 10 min. After demolding, the initial weights ($W_i$) of the freshly prepared samples were measured.

The gravimetric method was used to measure the swelling ratios of different protein hydrogels. The square-shaped hydrogels were first immersed in PBS and then subsequently in 0.5, 1.0, 1.5, 2.0, 2.5, 3.0, 4.0, 5.0, 6.0, and 7.0 M GdmCl

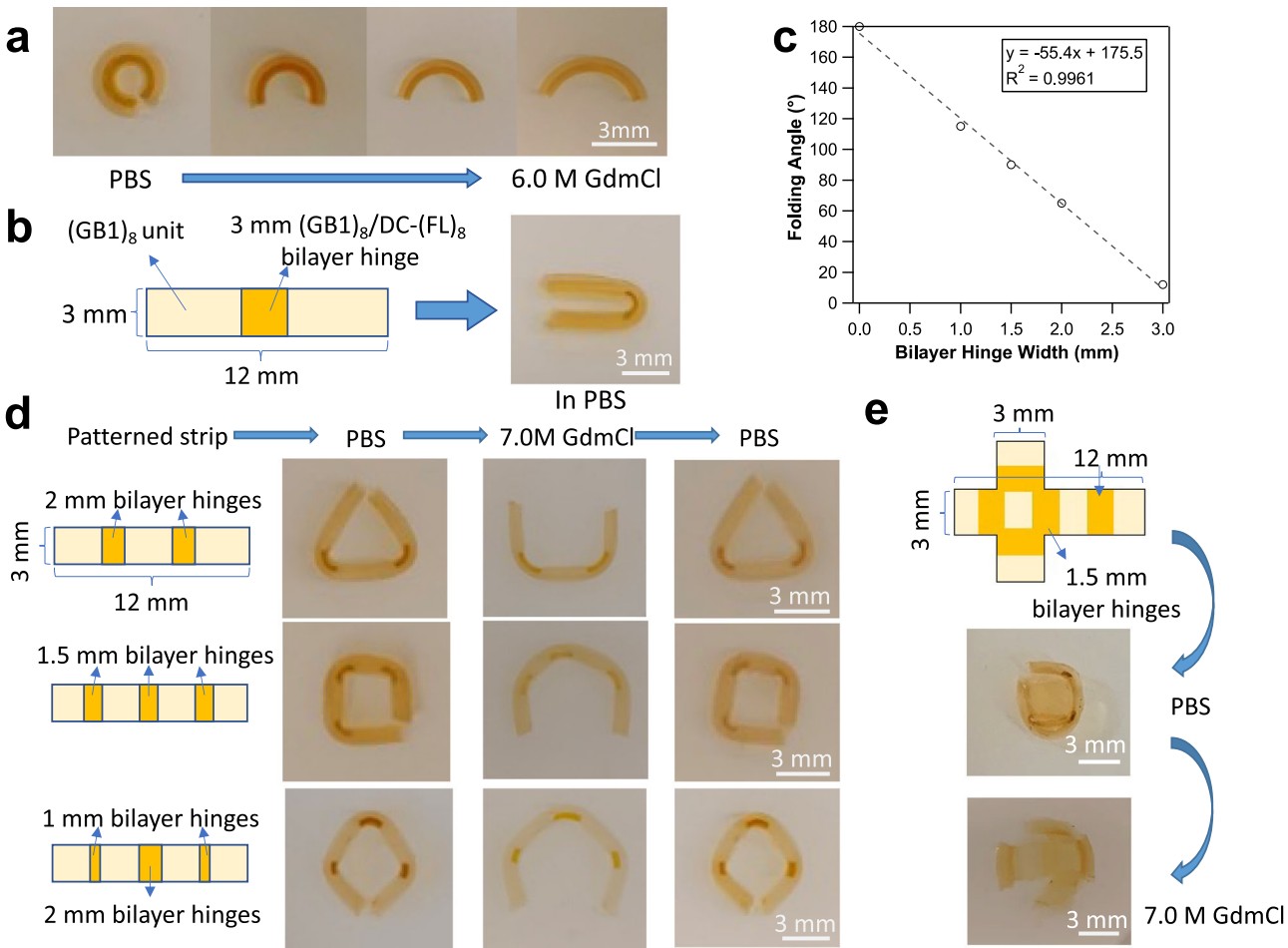

**Fig. 4 Complex hydrogel shape morphing can be programmed by using bilayer hinges. a** Arcs of different radii realized by soaking $(GB1)_8$/DC-$(FL)_8$ bilayer hydrogel in buffers at different [GdmCl]. **b** The folding deformation of a strip hydrogel patterned with a 3 mm $(GB1)_8$/DC-$(FL)_8$ bilayer hinge and two $(GB1)_8$ units. **c** The linear relationship between the strip folding angle and the width of $(GB1)_8$/DC-$(FL)_8$ hinge, which allows for precise programming of hydrogel folding deformation. This relationship was measured based on hydrogel strips with the following dimensions: length 12 mm, width 3 mm. **d** Recoverable 1D to 2D transformations of bilayer hinge patterned hydrogel strips to realize a triangle, a rectangle, and a rhombus shape. All the hydrogel strips are 12 mm in length and 3 mm in width. **e** 3D transformation to a cubic architecture from a $(GB1)_8$/DC-$(FL)_8$ bilayer hinge patterned 2D sheet.

solutions. Soaking was allowed for 3 h in each medium to achieve hydrogel swelling equilibrium. After each soaking step, the excess liquid on the hydrogel surface was wiped with Kimwipes and the hydrogel weight ($W_t$) was measured. The swelling ratio ($SR$) was calculated as

$$SR = \frac{W_t - W_i}{W_i} \times 100\% \qquad (2)$$

**Bilayer hydrogel preparation.** $(GB1)_8$/$(FL)_8$ bilayer hydrogels with layer thickness ratio of 1:1 were fabricated via layer-by-layer gelation using custom-made two-layer molds with a length of 10 mm, the width of 3.0 mm, and total depth of 1.2 mm or 0.8 mm (Supplementary Fig. 7a). Unless otherwise noted, both mold layers were of the same thickness. 20% $(FL)_8$ pre-gel solution was added into the first mold layer and preliminarily cross-linked under 200 W white light at 10 cm distance for 2 min. Then the second mold layer was attached right above the first mold layer, and 20% $(GB1)_8$ pre-gel solution was added. The mixture was irradiated using the same setup for 10 min. Bilayer hydrogels with different layer thickness ratios or protein content/network were prepared using the same method with adjusted individual mold layer thickness and protein pre-gel solution compositions, respectively. After demolding, the obtained bilayer hydrogels were immersed in PBS to reach swelling equilibrium.

For the patterned hydrogels with bilayer hinges, a custom-made two-layer mold with a length of 12 mm, width of 3.0 mm, and total depth of 0.8 mm was used (Supplementary Fig. 7b). DC-$(FL)_8$ layer was formed in the first mold layer and carved into the designed patterns within the mold. Then the second mold layer was attached, and $(GB1)_8$ pre-gel solution was added to fully fill the void. After gelated by 10 min white light irradiation, the patterned hydrogels were demolded and immersed in PBS to fully remove GdmCl salts and achieve swelling equilibrium.

**Bending test.** The bending behaviors of the as-prepared bilayer hydrogels were studied in PBS and in GdmCl solutions of various concentrations. Soaking was allowed for over 3 h in each medium to achieve hydrogel swelling equilibrium. Images of hydrogel bending status after each soaking step were taken. The bending angle, which is defined as the degree of deviation from the original linear position (Supplementary Fig. 8)[36], was measured using ImageJ.

**Instron test.** Tensile tests were used to characterize the mechanical properties of protein hydrogels following well-established protocols for soft hydrogel samples[46,47]. An Instron-5500R tensometer with a custom-made force gauge was used for the tensile tests. Young's modulus is derived from the stress-strain ratio from each stretching-relaxation curve at 15% strain.

## Data availability

The data that support the findings of this study are available from the corresponding author upon request. Source data are provided with this paper.

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

## Acknowledgements

This work is supported by the Natural Sciences and Engineering Research Council of Canada (NSERC). Q.B. acknowledges the fellowship support from NSERC NanoMat CREATE program.

## Author contributions

H.L. conceived the project. Q.B. and H.L. designed the experiments. Q.B. and L.F. performed the experiment. Q.B. analyzed the data. Q.B. and H.L. wrote the paper with contributions from L.F.

## Competing interests

The authors declare no competing interests.
