## [Peer Review File · Nature Communications]

REVIEWER COMMENTS

Reviewer #1 (Remarks to the Author):

The present manuscript demonstrates the engineering of protein-based shape memory/morphing hydrogels by utilizing protein folding-unfolding as a general mechanism to trigger shape-morphing in protein-bilayer structures. The authors used two different tandem modular elastomeric proteins (GB1)₈ and (FL)₈ as building blocks to successfully engineer a shape-memory/shape-morphing bilayer protein hydrogels due to different Young's moduli and swelling properties. The unfolding-folding induced changes in swelling results in a bidirectional bending deformation depending upon the buffer condition. Furthermore, the authors used the bilayer protein hydrogel as hinge to drive a folding deformation and realized 1D to 2D and 2D to 3D transformations of patterned hydrogels. Overall, the authors artfully combined the folding/unfolding of protein into the shape memory of biomaterials which is an exciting and interesting study for the broad audience, especially in the context of biomaterials, targeted by Nature Communications.

I can recommend its publication after addressing the following minor comments:

1. The authors claimed that "Once immersed in PBS, the bilayer strips spontaneously self-bentat equilibrium", but how quick would it get equilibrium and for the cycle tests? Does it take the same time to reach equilibrium in every cycle? Moreover, maybe the (GB1)₈ and (FL)₈ hydrogels reach equilibrium at different timescales.
2. Can the authors comments what is the molecular mechanism underlying distinct swelling behaviors of the two types protein hydrogels? Do cross-linking density, sequence hydrophobicity, or folding/unfolding rates play important roles?
3. The shape memory or the reversible character of hydrogel is related to the protein folding and unfolding, more references should be sited to support this assumption. Also, can the authors describe the role of mechanical properties of protein folding and unfolding in single molecule level in your hydrogel design, for example, the unfolding rate. Maybe coupled mechanical force and chemical denaturant can lead to distinct unfolding behaviors (e.g. Cao Y et al. J. Mol. Biol. 2008; Guinn EJ et al. J. Mol. Biol. 2017)
4. The authors should discuss whether the bending angles are affected by protein concentrations, cross-linking densities, thickness of each protein hydrogel layer, and buffer conditions. More importantly, they should provide a guideline to precisely control and regulate the bending angles.
5. Can the authors explain more clearly the roles of the swelling ratio and the modulus of the two hydrogels in the shape memory bilayer structure. The shape memory should be related to the interplay of the two effects. For example, the swelling of hydrogel A is much bigger than B while the Young's modulus is smaller, the bilayer structure might not bend. Maybe this is important for precisely engineering protein-based shape memory hydrogel. The modulus of these three hydrogels in different buffer should be provided.
6. The second colon in the legend of Figure 4C should be removed.
7. The number of samples should be reported.

Reviewer #2 (Remarks to the Author):

While unfolding-induced shape-memory in protein hydrogels based on interpenetrating networks was recently demonstrated (and published also by this journal), the current study builds one significant step forward by separating physically the two networks and producing a cantilever-type motion from the different unfolding and swelling responses of two model proteins. Similarly to a thermocouple which has two metals that respond differently to temperature, the authors intelligently use proteins that unfold at different chemical denaturant concentrations to produce shape memory (a mechano-couple?). While not mentioned in the discussion, in my opinion this is a simpler method, as one could use proteins without regard to charging and compatibility, and is a significant advancement, deserving full consideration for publication. I only have a few minor comments, which I think would further improve the manuscript.

1. Page 2 – “...allows precise control over identity” – what does this mean? Please explain.
2. In Fig.1 – I suggest to add a dotted vertical line marking the intersection point (0.6 M GuHCl), as this crossing plays an important role later in the design. I also think a new panel with the swelling ratio between the two proteins as a function of GuHCl would further strengthen the point.
3. Discuss swelling mechanisms, in particular why does the swelling for GB1 go down with GuHCl concentration, until 1 M.
4. In Eq. 1 the formula uses the stiffness ratio “ n ” (coming from Young’s moduli “ Y ” of the proteins forming hydrogels) in the absence of GuHCl. Since the authors have the tools and probably already measured the change in Y for both proteins as a function of chemical denaturant concentration, why not use those values instead? Would those $n(\text{GuHCl})$ better predict the bending angle? If yes, this would be also worthy of a Figure, if not, please discuss. As the swelling behavior is non-monotonic, I expect that Y s will not be as well. Furthermore, going back to my mechano-couple idea, $\Delta\alpha$ in this case should probably not be defined as “thermal” but “mechanical”/“swelling” expansion coefficients mismatch.
5. I commend the authors on the “local hinge” experiments, and I think if they also shot movies with these changes in shape, it will further add to the interest for this paper.

Many thanks for reviewing our manuscript (MS# NCOMMS-21-34682) entitled “Engineering Shape Memory/Morphing Protein Hydrogels Based on Protein Unfolding-Folding”. We thank both reviewers for their enthusiasm in our manuscript and constructive suggestions/comments. Following these comments/suggestions, we revised our manuscript accordingly. Changes we made to address reviewers’ comments are highlighted in yellow in the revised manuscript. We hope that our manuscript is now acceptable for publication in *Nature Communications*.

The response to the reviewer’s comments is detailed as follows:

Reviewer #1

“The present manuscript demonstrates the engineering of protein-based shape memory/morphing hydrogels by utilizing protein folding-unfolding as a general mechanism to trigger shape-morphing in protein-bilayer structures. The authors used two different tandem modular elastomeric proteins (GB1)₈ and (FL)₈ as building blocks to successfully engineer a shape-memory/shape-morphing bilayer protein hydrogels due to different Young’s moduli and swelling properties. The unfolding-folding induced changes in swelling results in a bidirectional bending deformation depending upon the buffer condition. Furthermore, the authors used the bilayer protein hydrogel as hinge to drive a folding deformation and realized 1D to 2D and 2D to 3D transformations of patterned hydrogels. Overall, the authors artfully combined the folding/unfolding of protein into the shape memory of biomaterials which is an exciting and interesting study for the broad audience, especially in the context of biomaterials, targeted by Nature Communications.”

Response: we thank this reviewer for his/her enthusiasm in our manuscript.

1. *“The authors claimed that “Once immersed in PBS, the bilayer strips spontaneously self-bentat equilibrium”, but how quick would it get equilibrium and for the cycle tests? Does it take the same time to reach equilibrium in every cycle? Moreover, maybe the (GB1)₈ and (FL)₈ hydrogels reach equilibrium at different timescales.”*

Response: following this comment, we have now provided kinetic information on the swelling process. Since the swelling/deswelling is mainly controlled by the diffusion of solvent in and out of protein hydrogels, swelling/deswelling are relatively slow processes. In addition, the kinetics of (GB1)₈ and (FL)₈ are similar. We have included these data as new Fig. S2 and S4 and discussions in the revised manuscript (see page 6).

2. *“Can the authors comments what is the molecular mechanism underlying distinct swelling behaviors of the two types protein hydrogels? Do cross-linking density, sequence hydrophobicity, or folding/unfolding rates play important roles?”*

Response: we appreciate this comment. The equilibrium-swelling properties of hydrogels reflect the combined effects of parameters including polymer-solvent interactions, polymer volume fraction, degree of crosslinking, system ionic character, etc, and can be described by the classical Flory-Rehner theory. Swelling equilibrium will be reached when the elastic free energy balances with the mixing free energy. Hence, crosslinking density and sequence hydrophobicity will affect the swelling ratio of protein hydrogels. The distinct swelling behaviors of the two protein hydrogels in PBS are due to the intrinsic mechanical stability of GB1 and FL. Single molecule force spectroscopy studies showed that GB1 is mechanically stable and requires ~180 pN to unfold while FL unfolds at ~5 pN at the same pulling speed. In (GB1)₈ hydrogels in PBS, GB1 domains remain folded, leading to a typical swelling behavior for protein hydrogels. However, due to the force generated by the swelling process is estimated to be around ~4-5 pN, a fraction of FL domains unfolded in the hydrogel upon swelling. Since PBS is a poor solvent, the unfolded FL polypeptide chain likely underwent hydrophobic collapse and formed aggregates, leading to the observed deswelling and opaque appearance. In the denatured state, the much larger swelling ratio of (FL)₈ is directly related to its lower Young's modulus, i.e. much smaller effective crosslinking density. We have included this discussion in the revised manuscript (see page 4).

3. *“The shape memory or the reversible character of hydrogel is related to the protein folding and unfolding, more references should be sited to support this assumption. Also, can the authors describe the role of mechanical properties of protein folding and unfolding in single molecule level in your hydrogel design, for example, the unfolding rate. Maybe coupled mechanical force and chemical denaturant can lead to distinct unfolding behaviors (e.g. Cao Y et al. J. Mol. Biol. 2008; Guinn EJ et al. J. Mol. Biol. 2017)”*

Response: it is worth noting that experimental efforts that use protein folding-unfolding as a means to engineer shape memory protein hydrogels have just begun. The only study prior to our current study is the work by Popa et al, who used protein unfolding-folding and interactions with charged polymers to engineer shape memory protein hydrogels.¹ We have made this point clear in the revised manuscript (see page 3). Since the reversible character of the bilayer hydrogel is due to the reversible folding-unfolding of GB1 and FL, we have cited three new references on the reversibility of folding-unfolding of GB1 and FL (see new ref. 34-35 and 37).

Mechanical properties of proteins certainly play important roles in determining the overall macroscopic mechanical properties of protein hydrogels, as shown in the example of GB1 and FL. However, rational design of protein hydrogels based on the mechanical properties of individual protein blocks remains challenging. As this reviewer correctly pointed out, coupled mechanical unfolding and chemical denaturation may lead to interesting behaviors of protein hydrogels. This will be worth exploring in future studies. We have now briefly commented on the mechano-chemical coupling in protein hydrogels in the revised manuscript (see page 4) and cited relevant papers (new ref. 34-36).

4. *“The authors should discuss whether the bending angles are affected by protein concentrations, cross-linking densities, thickness of each protein hydrogel layer, and buffer conditions. More importantly, they should provide a guideline to precisely control and regulate the bending angles.”*

Response: following this comment, we have included results showing the effect of protein concentrations (which is directly related to crosslinking density, new Fig. 2F), thickness of protein hydrogel layer (new Fig. S3) on the bending angles of the bilayer hydrogel strip. Based on these results, we have discussed design principles to control the bending angles of bilayer hydrogels (see page 9).

5. *“Can the authors explain more clearly the roles of the swelling ratio and the modulus of the two hydrogels in the shape memory bilayer structure. The shape memory should be related to the interplay of the two effects. For example, the swelling of hydrogel A is much bigger than B while the Young’s modulus is smaller, the bilayer structure might not bend. Maybe this is important for precisely engineering protein-based shape memory hydrogel. The modulus of these three hydrogels in different buffer should be provided.”*

Response: as this reviewer correctly pointed out, the Timoshenko’s theory states that the swelling ratio and the modulus of the two protein hydrogels (moduli ratio) directly determine the bending angle of the bilayer structure, as exemplified in Equ. 1. We have now emphasized this point in the revised manuscript (see page 9).

6. *“The second colon in the legend of Figure 4C should be removed.”*

Response: we have corrected this oversight.

7. *“The number of samples should be reported.”*

Response: we have now included this information in the revised manuscript.

Reviewer #2

“While unfolding-induced shape-memory in protein hydrogels based on interpenetrating networks was recently demonstrated (and published also by this journal), the current study builds one significant step forward by separating physically the two networks and producing a cantilever-type motion from the different unfolding and swelling responses of two model proteins. Similarly to a thermocouple which has two metals that respond differently to temperature, the authors intelligently use proteins that unfold at different chemical denaturant concentrations to produce shape memory (a mechano-couple?). While not mentioned in the discussion, in my opinion this is a simpler method, as one could use proteins without regard to charging and compatibility, and is a significant advancement, deserving full consideration for publication. I only have a few minor comments, which I think would further improve the manuscript.”

Response: we appreciate this reviewer's enthusiasm in our study.

1. *"Page 2 – "...allows precise control over identity" – what does this mean? Please explain."*

Response: "identify" was intended to refer to a specific protein. We regret the use of "identity" in this context, as it can cause confusion. We have deleted it in the revised manuscript.

2. *"In Fig. 1 – I suggest to add a dotted vertical line marking the intersection point (0.6 M GuHCl), as this crossing plays an important role later in the design. I also think a new panel with the swelling ratio between the two proteins as a function of GuHCl would further strengthen the point."*

Response: following this helpful suggestion, we have made the suggested change in Fig. 1. This reviewer may have confused Fig. 1C with Fig. S1, as Fig. 1C already shows the relationship between the swelling ratio of two proteins and [GuHCl].

3. *"Discuss swelling mechanisms, in particular why does the swelling for GB1 go down with GuHCl concentration, until 1 M."*

Response: the slight decrease in the swelling degree of (GB1)₈ hydrogel when it was transferred to 0.5 M GdmCl solution was possibly a result of the ionic strength change in the aqueous environment. At 0 M [GdmCl], the hydrogel was soaked in PBS buffer at ionic strength of ~0.3 M. After transferred to 0.5 M GdmCl, the increase in ionic strength could lead to the decrease in hydrogel swelling degree. We have included this discussion in the revised manuscript (see page 5).

4. *"In Eq. 1 the formula uses the stiffness ratio "n" (coming from Young's moduli "Y" of the proteins forming hydrogels) in the absence of GuHCl. Since the authors have the tools and probably already measured the change in Y for both proteins as a function of chemical denaturant concentration, why not use those values instead? Would those n(GuHCl) better predict the bending angle? If yes, this would be also worthy of a Figure, if not, please discuss. As the swelling behavior is non-monotonic, I expect that Ys will not be as well. Furthermore, going back to my mechano-couple idea, delta-alpha in this case should probably not be defined as "thermal" but "mechanical"/"swelling" expansion coefficients mismatch."*

Response: Equation 1 in the text is the equation of the Timoshenko's bilayer theory which was derived for bi-metallic strips. To better describe bilayer hydrogel strips, the $\Delta\alpha$ should be defined as the mismatch between the swelling equilibrium ratios of the two hydrogel layers.^{2,3} We have revised this point in the revised manuscript (see page 8). It is important to note that quantitative prediction of bending angles of bilayer hydrogels remains challenging. And in this work all the comparisons between our experiments and theory are qualitatively in nature. We have emphasized this qualitative nature of the present work in the revised manuscript (see page 8-9).

5. *"I commend the authors on the "local hinge" experiments, and I think if they also shot movies with these changes in shape, it will further add to the interest for this paper."*

Response: we appreciate this comment. Following this suggestion, we have included two movies in Supplementary Information to show the folding/unfolding process of a triangle (see supplementary movie 1 and 2). In addition, we also included morphing kinetics of bilayer hydrogel strips as new Fig. S2 and S4.

Again, we thank both reviewers for their constructive comments and suggestions. We hope that the revised manuscript is now acceptable for publication in *Nature Communications*.

Sincerely yours,

Hongbin Li, Ph. D.
Professor
Department of Chemistry
University of British Columbia
Vancouver, BC V6T 1Z1
Canada
Tel: 604-822-9669
Email: hongbin@chem.ubc.ca

References

- 1 Khoury, L. R. & Popa, I. Chemical unfolding of protein domains induces shape change in programmed protein hydrogels. *Nat. Comm.* **10**, 1-9, (2019).
- 2 Lucantonio, A., Nardinocchi, P. & Pezulla, M. Swelling-induced and controlled curving in layered gel beams. *Proc Math Phys Eng Sci* **470**, 20140467, (2014).
- 3 Egunov, A. I., Korvink, J. G. & Luchnikov, V. A. Polydimethylsiloxane bilayer films with an embedded spontaneous curvature. *Soft Matter* **12**, 45-52, (2015).

REVIEWERS' COMMENTS

Reviewer #1 (Remarks to the Author):

The authors have done an excellent job in revision. My comments have been adequately addressed.

I am just curious: The authors suggest that the bilayer curvature can be predicted using the Timoshenko's bilayer theory (Equation 1). It seems that the authors have all the numbers for the variables in this equation. How do the predicted bending angles differ from the experimentally measured ones? The authors may have the opportunity to test whether indeed this theory originally developed for bilayer metal strips can be extended to soft hydrogels.

Additionally, a few minor comments.

Number of samples is not reported in Figure 2C.

"%" is missing for the "20%" label in Figure 2F.

A scale bar may be needed for each photograph.

Some recent papers about protein based shape-memory hydrogels may need to be cited.

"Elastin-Based Thermo-responsive Shape-Memory Hydrogels" *Biomacromolecules* 2020, 21, 3, 1149–1156

"Salt-Induced Shape-Memory Effect in Gelatin-Based Hydrogels"
Biomacromolecules 2020, 21, 6, 2024–2031

"Thermally-Induced Shape-Memory Behavior of Degradable Gelatin-Based Networks"
Int. J. Mol. Sci. 2021, 22(11), 5892

The paper does not need to return to me for further comments.

Reviewer #2 (Remarks to the Author):

I am happy with the implemented changes.

Many thanks for reviewing our manuscript (MS# NCOMMS-21-34682) entitled “Engineering Shape Memory/Morphing Protein Hydrogels Based on Protein Unfolding-Folding”. Following Reviewer#1’s comments, as well as your editorial comments, we have revised our manuscript accordingly. Changes we made are highlighted in yellow in the revised manuscript. We hope that our manuscript can now be accepted for publication in *Nature Communications*.

The response to the reviewer’s comments is detailed as follows:

Reviewer #1

1. *“I am just curious: The authors suggest that the bilayer curvature can be predicted using the Timoshenko’s bilayer theory (Equation 1). It seems that the authors have all the numbers for the variables in this equation. How do the predicted bending angles differ from the experimentally measured ones? The authors may have the opportunity to test whether indeed this theory originally developed for bilayer metal strips can be extended to soft hydrogels.”*

Response: similar to previous studies,¹⁻³ our results showed that the Timoshenko’s bilayer theory can be used to qualitatively predict the behaviors of the protein bilayer hydrogels, but a quantitative agreement between experiments and theory is still lacking. We have made this point clear in the revised manuscript (see page...).

2. *“Additionally, a few minor comments.
Number of samples is not reported in Figure 2C.
“%” is missing for the “20%” label in Figure 2F.
A scale bar may be needed for each photograph.”*

Response: we have now included the missing information in the revised manuscript.

3. *“Some recent papers about protein based shape-memory hydrogels may need to be cited.
“Elastin-Based Thermoresponsive Shape-Memory Hydrogels” *Biomacromolecules* 2020, 21, 3, 1149–1156
“Salt-Induced Shape-Memory Effect in Gelatin-Based Hydrogels”
Biomacromolecules 2020, 21, 6, 2024–2031
“Thermally-Induced Shape-Memory Behavior of Degradable Gelatin-Based Networks”
Int. J. Mol. Sci. 2021, 22(11), 5892”*

Response: we thank this reviewer for bringing these papers to our attention. We have now cited these references.

We hope that the revised manuscript can now be accepted for publication in *Nature Communications*.

Sincerely yours,

Hongbin Li, Ph. D.
Professor
Department of Chemistry
University of British Columbia
Vancouver, BC V6T 1Z1
Canada
Tel: 604-822-9669
Email: hongbin@chem.ubc.ca

References

- 1 Egunov, A. I., Korvink, J. G. & Luchnikov, V. A. Polydimethylsiloxane bilayer films with an embedded spontaneous curvature. *Soft Matter* **12**, 45-52, (2015).
- 2 Ryu, J., D'Amato, M., Cui, X., Long, K. N., Qi, J., Dunn, M. L. . Photo-origami—Bending and folding polymers with light. *Appl. Phys. Lett.* **100**, 161908, (2012).
- 3 van Manen, T., Janbaz, S. & Zadpoor, A. A. Programming the shape-shifting of flat soft matter. *Mater. Today* **21**, 144-163, (2018).